# Elevated Uric Acid Levels with Early Chronic Kidney Disease as an Indicator of New-Onset Ischemic Heart Disease: A Cohort of Koreans without Diabetes

**DOI:** 10.3390/biomedicines11082212

**Published:** 2023-08-07

**Authors:** Sung-Bum Lee, Hui-Jeong Lee, Ha Eun Ryu, Byoungjin Park, Dong-Hyuk Jung

**Affiliations:** 1Department of Family Medicine, Soonchunhyang University Bucheon Hospital, Bucheon-si 22972, Republic of Korea; sblee@schmc.ac.kr (S.-B.L.); hjlee@schmc.ac.kr (H.-J.L.); 2Department of Family Medicine, Yonsei University College of Medicine, Seoul 03722, Republic of Korea; rahahaha@yuhs.ac

**Keywords:** chronic kidney disease, uric acid, ischemic heart disease, cohort study

## Abstract

Several studies have showed that hyperuricemia is related to the development of ischemic heart disease (IHD). There is also growing evidence indicating that hyperuricemia may contribute to the progression of IHD as a pathogenic factor. Ironically, uric acid can be an antioxidant agent, as shown in experimental studies. The aim of our study is to analyse the association between uric acid and IHD with early-stage chronic kidney disease (CKD). Data were assessed from 17,492 participants without cardiovascular disease from the Korean Genome and Epidemiology Study (KoGES) and Korea Health Insurance Review and Assessment (HIRA) data. The subjects were categorized as four groups according to CKD and uric acid levels. We retrospectively evaluated hazard ratios (HRs) with 95% confidence intervals (CIs) for IHD by using multivariate Cox regression analysis over a 4-year period from the baseline survey. During the follow-up, 335 individuals (3.4%; 236 men and 99 women) developed IHD. Compared to the participants without elevated uric acid and early CKD HRs for incident IHD according to uric acid levels and early CKD, the uric acid level was 1.13 (95% CI, 0.86–1.48) in participants with elevated uric acid and without early CKD, 0.99 (95% CI, 0.55–1.77) in participants without elevated uric acid and with early CKD, and 1.65 (95% CI, 1.03–2.66) in participants with elevated uric acid and early CKD after adjusting for confounding metabolic factors. Early CKD and high uric acid levels increased the risk of new-onset IHD (HR, 1.65; 95% CI, 1.03–2.66). Elevated uric acid levels were related to an increased risk of incident IHD in early-stage CKD patients. It is expected that uric acid can be a reliable predictor for IHD, even in early-stage CKD patients; thus, in those with CKD, proactively managing uric acid levels can play a significant role in reducing the risk of cardiovascular disease.

## 1. Introduction

The prevalence of hyperuricemia has been on the rise over the decades; concomitantly, the burden of managing the disease is progressively growing. A study in the United Kingdom found that the overall prevalence of hyperuricemia had risen from 0.26% to 0.95% between 1970 and 1993 [1]. Furthermore, another study in the United States showed the prevalence of gout in elderly older than 75 years old had doubled from 2.1% to 4.1% between 1990 and 1999 [2]. Gout, which is clinical condition after sustained hyperuricemia, is estimated to impact 5.1 million people in the U.S., according to the Third National Health and Nutrition Survey (NHANES III) [3]. Furthermore, high uric acid levels are recognized as a risk factor of hypertension, diabetes, and metabolic syndrome, which are known risk factors for cardiovascular diseases (CVDs) [4,5,6]. Nevertheless, the mechanism and causality of the relationship between uric acid and CVDs are controversial because uric acid is strongly collinear with metabolic syndrome [7].

The early identification and correction of hyperuricemia is an important medical issue that has to be addressed before the progression to comorbidity that induces a range of complications [8]. However, many people are inclined to overlook elevated uric acid because individuals are often asymptomatic before progressing to gout.

Early CKD is an important public health problem even before progression to advanced stages and ultimately ESRD, which can cause a series of complications such as anaemia and bone and mineral metabolism disorder [9]. Nevertheless, it is easy to neglect early CKD and hyperuricemia as they are often asymptomatic. In NHANES III, only 8% of patients knew about their condition [10].

Hyperuricemia is more common in patients with CKD than in those without CKD because hyperuricemia and CKD share common risk factors: obesity, hypertension, dyslipidemia, and diabetes [11,12]. In spite of the potential link between uric acid and early CKD regarding ischemic heart disease (IHD), few studies have shown a causal relationship between cardiovascular risks and uric acid and CKD because of the similarity in well-known risk factors. Consequently, we retrospectively investigated the combined impact of increased uric acid levels and early CKD on new-onset IHD risk in a large-scale cohort of Koreans without diabetes.

## 2. Materials and Methods

### 2.1. Study Design and Participants

This study aimed to investigate surrogate indicators for IHD among Korean adults using data from the Health Insurance Review and Assessment Service (HERAS-HIRA dataset) based on health-risk-assessment studies. The cohort data included 20,530 participants who were sequentially enrolled during their voluntary visits to the Health Promotion Centre at Gangnam Severance Hospital, Yonsei University College of Medicine, for a health check-up between November 2006 and June 2010. After the initial assessment of subjects, we excluded 1590 (7.7%) individuals who had a history of IHD or stroke, previously diagnosed type 2 diabetes, or a fasting plasma glucose (FPG) level equal to or greater than 126 mg/dL. Participants meeting any of the following criteria were also excluded from the study: age younger than 20 years, C-reactive protein (CRP) level ≥ 5.0 mg/L, chronic kidney disease G3b-5, missing data, or currently taking aspirin (*n* = 1448). After applying these exclusion criteria, the final analysis included 17,492 participants (8822 men and 8670 women), as shown in Figure 1. This study was performed in accordance with the ethical principles of the Declaration of Helsinki and was approved by the Institutional Review Board (IRB) of Yonsei University College of Medicine, Seoul, Korea (reference number: 2015-32-0009). The HERAS and HIRA datasets which were used in our study are available from the corresponding author upon reasonable request.

### 2.2. Data Collection

All participants completed a comprehensive questionnaire regarding their lifestyle and medical history. Information on self-reported cigarette smoking, alcohol consumption, and physical activity was collected from these questionnaires. Smoking status was classified as never smoker, ex-smoker, or current smoker. Regular alcohol intake was defined as consuming ≥140 g of alcohol per week [13]. Participants provided information about their level of physical activity, and regular exercise was regarded as physical activity of either moderate or vigorous intensity more than or equal three times per week [14]. Systolic and diastolic blood pressure was measured on the participant’s right arm by using a standard mercury sphygmomanometer while they were in a seated position after a 10 min rest (Baumanometer, W.A. Baum Co., Inc., Copiague, NY, USA). Blood samples were obtained from the antecubital vein after a 12 h overnight fast, and total cholesterol, triglyceride, high-density lipoprotein cholesterol, and liver enzymes were measured by enzymatic methods using a Hitachi 7600 automated chemistry analyser (Hitachi Co., Tokyo, Japan). The CRP concentrations were collected with a Roche/Hitachi 912 System (Roche Diagnostics, Indianapolis, IN, USA) using a latex-enhanced immunoturbidimetric method. Hypertension was defined as a systolic blood pressure of more than or equal 140 mmHg, diastolic blood pressure of more than or equal 90 mmHg, or the current use of anti-hypertensive medication [15].

### 2.3. Chronic Kidney Disease

Chronic kidney disease (CKD) was defined as an estimated glomerular filtration rate (eGFR) lower than 60 mL/min/1.73 m^2^. The eGFR was calculated utilizing the abbreviated equation derived from the Modification of Diet in Renal Disease (MDRD) study: 186.3 × serum creatinine^−1.154^ × age^−0.203^ × 0.742 (if a woman) or proteinuria ≥ 1+ [16]. CKD stage 1 was defined when the eGFR was 90 or more and proteinuria of 1 or more. CKD stage 2 was defined as cases with an eGFR between 60 and less than 90 and proteinuria of 1 or more. CKD stage 3a was defined as cases with an eGFR between 45 and less than 60 regardless of the presence of proteinuria [17].

### 2.4. Study Outcomes

The study focused on evaluating the outcome of ischemic heart disease (IHD), which encompassed angina (ICD-10 code I20) or acute myocardial infarction (ICD-10 code I21) occurring after the participants’ initial enrolment in the study. To establish baseline and follow-up survey outcomes, we connected each subject’s personal identification number with the available HIRA data in South Korea from November 2006 to December 2010.

### 2.5. Statistical Analysis

We classified the participants into uric acid control and elevated uric acid groups based on the 75th percentile with a threshold of more than or equal 6.5 mg/dL in men and more than or equal 4.6 mg/dL in women, respectively [18]. Then, we subdivided the study participants into four groups: no early CKD and control of uric acid (Group 1), no early CKD and high uric acid (Group 2), early CKD and control of uric acid (Group 3), and early CKD and elevated uric acid (Group 4). Analysis of variance (ANOVA) was used for continuous variables to compare the baseline characteristics of the participants in each group, while the chi-square test was utilised for categorical variables. Age- and sex-adjusted survival curves were employed to compute the cumulative incidence of ischemic heart disease (IHD) for each group. Regarding Group 1 as the reference group, we calculated hazard ratios (HRs) and the corresponding 95% confidence intervals (CIs) for IHD through multivariate Cox proportional hazards regression analysis. These models were adjusted for potential confounding variables. In order to assess the combined effect, we conducted a comparative analysis of the hazard ratio (HR) considering the presence of each factor individually and in combination among all participants. All statistical analyses were conducted using SAS version 9.4 (SAS Institute Inc., Cary, NC, USA). All statistical tests were two-sided, and *p*-values less than 0.05 were regarded as statistically significant.

## 3. Results

Table 1 suggests the baseline characteristics of the participants according to early CKD and elevated uric acid. A total of 17,492 participants were finally included in our study. There are a total of 16,452 participants without CKD and 1040 participants with early CKD. The average age and BMI of the overall study population were 44.8 ± 10.5 years and 23.2 ± 3.0 kg/m^2^, respectively. The average uric acid level of the participants was 5.0 ± 1.4 mg/dL, and the average of estimated glomerular filtration rate (eGFR) of the overall participants was 83.8 ± 13.5 mL/min/1.73 m^2^. The subjects were subdivided into two groups according to the existence of early CKD. The percentages of early CKD stage 1-3a prevalence were, respectively, categorized as follows: 0.9% for stage 1, 3.2% for stage 2, and 1.9% for stage 3a. Regardless of the presence of early CKD, significant differences were observed based on uric acid levels. The high uric acid group showed higher levels of the male sex, BMI, systolic blood pressure, diastolic blood pressure, FPG, total cholesterol, triglyceride, and CRP compared to the control group (*p* < 0.001). Conversely, the elevated uric acid groups had lower levels of high-density lipoprotein (HDL) cholesterol and eGFR values (*p* < 0.001). Lifestyle factors such as smoking and alcohol drinking exhibited significant differences among groups (*p* < 0.001), while regular exercise did not differ significantly (*p* = 0.861). Furthermore, the elevated uric acid groups had a higher prevalence of hypertension and impaired fasting glucose compared to the control group (*p* < 0.001).

Table 2 suggests the results of the multivariate Cox proportional hazards regression analysis for the prediction of new-onset IHD according to uric acid levels and early CKD. A total of 335 individuals (3.4%) incurred IHD during the follow-up period. The incidence rate (per 1000 people years) of IHD showed a clear increasing trend, with higher rates observed in individuals with early CKD and elevated uric acid levels. Notably, in Model 3, which accounted for various covariates including smoking history, alcohol history, physical activity, blood pressure, fasting plasma glucose, total cholesterol, hypertension, and dyslipidemia, the elevated uric acid group with early CKD had a significantly higher hazard ratio of 1.65 (95% CI: 1.03–2.66) compared to the control group with no early CKD, indicating an increased risk of developing IHD. On the other hand, HRs (95% CI) for incident IHD were 1.13 (0.86–1.48) for the elevated uric acid group without CKD and 0.99 (0.55–1.77) for the control uric acid group with early CKD, which were statistically insignificant.

Multivariate Cox proportional hazards regression analysis was also conducted for the prediction of IHD according to early CKD, high uric acid, and their combination, as shown in Table 3. In Model 3, accounting for various potentially confounding variables, it showed statistical insignificance in the incidence of IHD between the CKD and elevated uric acid groups when compared to their respective control groups. Figure 2 illustrates the incidence of IHD across four groups distinguished by the control or elevation of uric acid levels and the presence or absence of early CKD. The incidence of IHD was highest in Group 4 (elevated uric acid with early CKD), followed by Group 3 (controlled uric acid with early CKD). Group 2 (elevated uric acid without CKD) had a lower incidence of IHD compared to the previous groups but a higher incidence than Group 1 (control of uric acid without CKD).

## 4. Discussion

In these large retrospective cohort data, high uric acid levels with early CKD could predict incident IHD among individuals without diabetes. Uric acid was an independent indicator of IHD irrespective of age, sex, body mass index, smoking status, alcohol intake, and physical activity.

The association of uric acid with cardiovascular disease is widely recognized. Several studies have found the relationship between uric acid and cardiovascular disease [19,20]. However, it has not been definitely proposed if uric acid is an indicator for risk in cardiovascular disease or if treatment lowering uric acid affects outcomes. Although uric acid has an independent correlation with cardiovascular risk, Dobson et al. argued that the association was due to uric acid being strongly collinear with traditional cardiovascular risk (i.e., metabolic syndrome) [7]. The other study showed that pathophysiology and outcomes for the association of uric acid with cardiovascular disease remain controversial [21]. However, we found a relationship between elevated uric acid and CKD and IHD in a large number of subjects over 4 years of follow-up. Furthermore, the association still remained significant despite adjusting for mean arterial blood pressure, fasting plasma glucose, total cholesterol, hypertension medication, and dyslipidemia medication, which are the primary determinants of metabolic syndrome. Specifically, we revealed that the uric acid level in early CKD patients was significantly associated with IHD, although the association of uric acid with IHD remained insignificant.

Uric acid could serve as an indicator for the development of both metabolic syndrome and cardiovascular disease associated with oxidative stress, although uric acid is known to be a major antioxidant in human plasma [22,23]. It is known that uric acid can scavenge oxygen radicals and protect the erythrocyte membrane from lipid oxidation [24]. However, the impact of uric acid was observed under particular circumstances in which exogenously introduced uric acid protected cells from oxidants. These oxidants were also added externally to the aqueous incubation media. Furthermore, even the plasma urate can inhibit lipid peroxidation only as long as ascorbic acid is present [25]. Paradoxically, uric acid might function either as an antioxidant (in plasma) or pro-oxidant (within the cell) [22]. The hydrophobic environment induced by lipids is adverse to the antioxidant effects of uric acid [26], and event-oxidised lipids can change uric acid into an oxidant [27]. Furthermore, many studies address that the entry of uric acid into cells is related to a decrease in NO bioavailability [28,29], which can induce endothelial dysfunction. In other words, increased serum uric acid has also been shown to cause arteriolar disease in the kidney, which can cause CKD [30]. In turn, increased levels of serum uric acid have been consistently recognized as a powerful independent indicator of hypertension in many studies [31,32,33]. Moreover, the occurrence of hypertension is avoidable by lowering uric acid levels [34]. Accordingly, uric acid increases the risk of IHD because of its contribution to hypertension [35]. Even though it is demanding to prove whether elevated uric acid and early CKD synergistically contribute to the development of IHD, it can be inferred that the patients with hyperuricemia and early CKD have been exposed to prolonged exposure to high levels of uric acid.

In spite of the extensive cohort study, there were several limitations to our research. First, this study did not include the effect of changes in uric acid levels and CKD stage, which could impact the new-onset IHD during the follow-up period. Second, the types of hypertensive medication were not considered (e.g., calcium channel blocker, angiotensin-converting enzyme inhibitor, and angiotensin receptor blocker). The variables which could differentially affect IHD were not perfectly adjusted for in the statistical models [36]. Furthermore, there are no data available on anti-hyperuricemic medication history, which could affect uric acid level. Fourth, the urine dipstick test is a qualitative test unlike 24 h urine collection. Furthermore, because there are no data for the urinary albumin-to-creatinine ratio, it can miss albuminuria which is a determinant of CKD. Finally, the study population did not exactly reflect the overall Korean population. Individuals who participated in health check-up programs are commonly concerned about their health issues, leading to a selection bias.

## 5. Conclusions

In summary, increased uric acid levels could be used to estimate the occurrence of IHD in patients with CKD in stages even earlier than G3a. The appropriate detection of both uric acid levels and CKD, even in the earlier stages, is essential to deter the development of IHD. In other words, if the uric acid level is 6.5 mg/dL or higher in a male with early CKD, and 4.6 mg/dL or higher in a female with early CKD, proper management is recommended in order to inhibit the occurrence of IHD. Further studies are needed to clarify the direct association of uric acid with CKD.

## Figures and Tables

**Figure 1 biomedicines-11-02212-f001:**
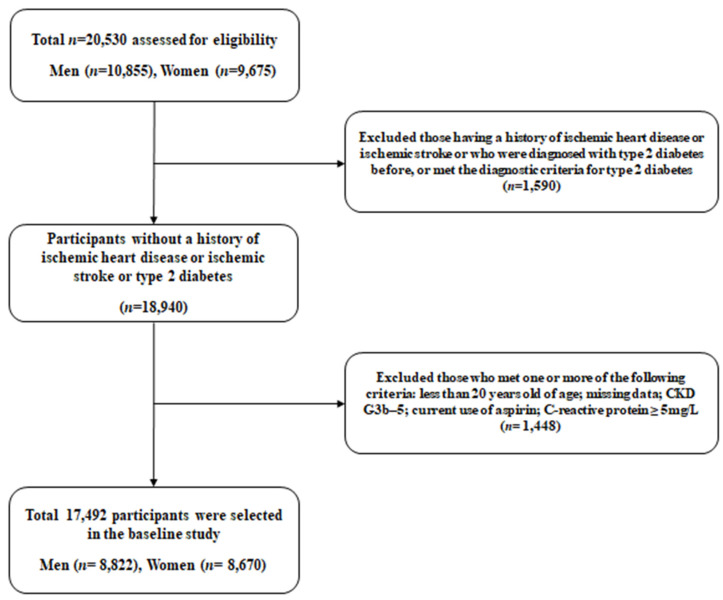
Flow diagram meeting the criteria for the selection of study participants. Abbreviations: CKD, chronic kidney disease.

**Figure 2 biomedicines-11-02212-f002:**
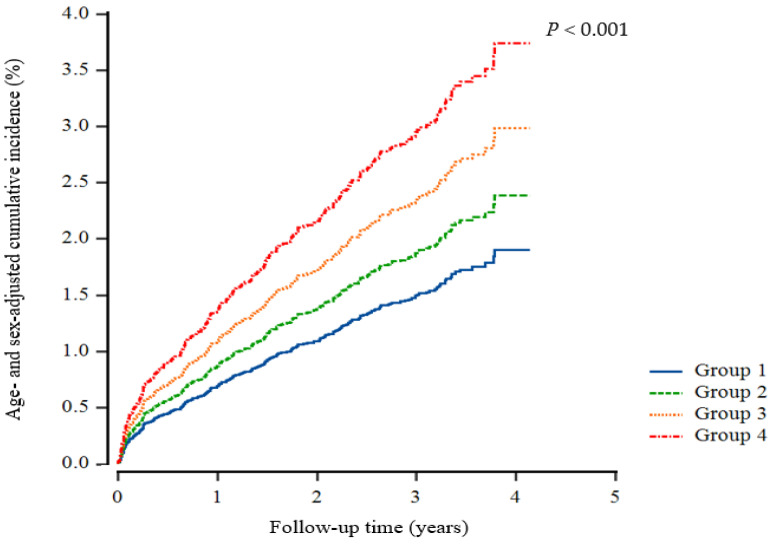
Cox regression cumulative incidence curve for ischemic heart disease. Group 1, control of uric acid without CKD; Group 2, elevated uric acid without CKD; Group 3, control of uric acid with early CKD; Group 4, elevated uric acid with early CKD.

**Table 1 biomedicines-11-02212-t001:** Baseline characteristics of the study population.

Variables	Overall	No CKD		Early CKD (G1-G3a)		*p* Value ^a^	Post Hoc ^b^
		Control of Uric Acid	Elevated Uric Acid	Control of Uric Acid	Elevated Uric Acid		
Number of participants, *n*	17,492	12,257	4,195	607	433		
Age (years)	44.8 ± 10.5	44.7 ± 10.3	44.5 ± 10.8	46.9 ± 11.8	48.1 ± 12.1	<0.001	b,c,d,e
Male sex (%)	50.4	49.5	51.4	57.7	58.4	<0.001	-
BMI (kg/m^2^)	23.2 ± 3.0	22.9 ± 2.9	24.2 ± 3.2	22.8 ± 3.1	24.2 ± 3.1	<0.001	a,c,d,f
Systolic BP (mmHg)	121.6 ± 15.4	120.4 ± 15.2	124.1 ± 15.6	123.3 ± 16.2	127.2 ± 16.1	<0.001	a,b,c,d,e,f
Diastolic BP (mmHg)	75.8 ± 10.1	75.1 ± 9.9	77.6 ± 10.1	77.0 ± 10.5	79.7 ± 10.2	<0.001	a,b,c,e,f
FPG (mg/dL)	91.0 ± 9.7	90.5 ± 9.3	92.4 ± 10.3	91.7 ± 10.9	94.0 ± 11.5	<0.001	a,b,c,e,f
Total cholesterol (mg/dL)	189.0 ± 33.5	185.8 ± 32.4	196.5 ± 34.6	193.5 ± 34.4	201.5 ± 36.7	<0.001	a,b,c,e,f
Triglyceride (mg/dL)	122.6 ± 84.9	113.8 ± 78.7	145.0 ± 95.5	121.7 ± 86.9	156.7 ± 94.0	<0.001	a,b,c,e,f
HDL-cholesterol (mg/dL)	53.7 ± 12.7	54.4 ± 12.7	51.8 ± 12.4	54.5 ± 14.0	51.4 ± 13.9	<0.001	a,c,d,f
C-reactive protein (mg/L)	0.9 ± 0.9	0.8 ± 0.8	1.1 ± 1.0	1.0 ± 1.0	1.2 ± 1.1	<0.001	a,b,c,d,e,f
Uric acid (mg/dL)	5.0 ± 1.4	4.5 ± 1.1	6.2 ± 1.2	4.7 ± 1.0	6.6 ± 1.3	<0.001	a,b,c,d,e,f
eGFR (mL/min/1.73 m^2^)	83.8 ± 13.5	85.5 ± 13.2	81.1 ± 11.9	76.9 ± 15.7	69.0 ± 14.5	<0.001	a,b,c,d,e,f
Current smoker (%)	24.4	23.0	27.6	28.7	26.0	<0.001	-
Alcohol drinking (%) ^c^	43.6	42.8	45.4	42.9	50.2	<0.001	-
Regular exercise (%) ^d^	30.5	30.7	30.0	31.3	30.5	0.861	-
Impaired fasting glucose (%)	17.1	14.9	21.7	19.8	30.0	<0.001	-
Hypertension (%)	19.8	17.0	25.7	24.1	38.1	<0.001	-

Note: BMI, body mass index; BP, blood pressure; FPG, fasting plasma glucose; HDL, high-density lipoprotein; eGFR, estimated glomerular filtration rate. ^a^ *p* values were calculated using one-way ANOVA or Pearson’s chi-square test. ^b^ Post hoc analysis with the Bonferroni method for mean differences between groups: a, Group 1 versus Group 2; b, Group 1 versus Group 3, c: Group 1 versus Group 4; d, Group 2 versus Group 3; e, Group 2 versus Group 4, and f, Group 3 versus Group 4. ^c^ Alcohol intake ≥ 140 g/week. ^d^ Moderate-intensity physical exercise ≥ three times/week.

**Table 2 biomedicines-11-02212-t002:** Hazard ratios and 95% confidence intervals for new-onset ischemic heart diseases.

		No Early CKD		Early CKD (G1-G3a)		*p* for Trend
		Control of Uric Acid	Elevated Uric Acid	Control of Uric Acid	Elevated Uric Acid	
New cases of ischemic heart disease, *n*		209	88	16	22	
Mean follow-up, years		2.4 ± 1.1	2.3 ± 1.1	2.3 ± 0.9	2.3 ± 1.0	
Pearson years of follow-up		29,111	9808	1393	1008	
Incidence rate/1000 person years		7.2	9.0	11.5	21.8	
Model 1	HR (95% CI)	1.00	1.28 (0.99–1.64)	1.24 (0.75–2.07)	2.22 (1.42–3.45)	0.002
	*p* value		0.052	0.406	<0.001	
Model 2	HR (95% CI)	1.00	1.26 (0.97–1.65)	1.03 (0.58–1.85)	2.10 (1.32–3.35)	0.011
	*p* value		0.088	0.915	0.001	
Model 3	HR (95% CI)	1.00	1.13 (0.86–1.48)	0.99 (0.55–1.77)	1.65 (1.03–2.66)	0.205
	*p* value		0.378	0.959	0.038	

Model 1: adjusted for age and sex. Model 2: adjusted for age, sex, body mass index, smoking status, alcohol intake, and physical activity. Model 3: adjusted for age, sex, body mass index, smoking status, alcohol intake, physical activity, mean arterial blood pressure, fasting plasma glucose, total cholesterol, hypertension medication, and dyslipidemia medication.

**Table 3 biomedicines-11-02212-t003:** Hazard ratios and 95% confidence intervals for new-onset ischemic heart disease according to early CKD, high uric acid, and their combination.

		No Early CKD	Early CKD	Control of Uric Acid	Elevated Uric Acid	Control of Uric Acid with No Early CKD	Elevated Uric Acid with Early CKD
New cases of ischemic heart disease, *n*		297	38	225	110	209	22
Mean follow-up, years		2.4 ± 1.1	2.3 ± 0.9	2.4 ± 1.1	2.3 ± 1.1	2.4 ± 1.1	2.3 ± 1.0
Pearson years of follow-up		38,919	2401	30.504	10,816	29,111	1008
Incidence rate/1000 person years		7.6	15.8	7.4	10.2	7.2	21.8
Model 1	HR (95% CI)	1.00	1.56 (1.11–2.19)	1.00	1.38 (1.10–1.73)	1.00	2.22 (1.42–3.45)
	*p* value		0.010		0.005		<0.001
Model 2	HR (95% CI)	1.00	1.42 (0.98–2.05)	1.00	1.37 (1.07–1.75)	1.00	2.10 (1.32–3.35)
	*p* value		0.063		0.012		0.001
Model 3	HR (95% CI)	1.00	1.26 (0.87–1.83)	1.00	1.20 (0.94–1.55)	1.00	1.65 (1.03–2.66)
	*p* value		0.219		0.148		0.038

Model 1: adjusted for age and sex. Model 2: adjusted for age, sex, body mass index, smoking status, alcohol intake, and physical activity. Model 3: adjusted for age, sex, body mass index, smoking status, alcohol intake, physical activity, mean arterial blood pressure, fasting plasma glucose, total cholesterol, hypertension medication, and dyslipidemia medication.

## Data Availability

The datasets used and analysed in the current study are available from the corresponding author on reasonable request.

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
