# Peer review of "Elevated Uric Acid Levels with Early Chronic Kidney Disease as an Indicator of New-Onset Ischemic Heart Disease: A Cohort of Koreans without Diabetes"

_biomedicines, 2023, doi:10.3390/biomedicines11082212_

Round 1

Reviewer 1 Report

Major revision is required. 

Methods:

1. The definition of CKD seems to be rough. According to KDIGO guideline, CKD is defined as kidney damage or glomerular filtration rate (GFR) <60 mL/min/1.73 m2 for 3 months or more, irrespective of cause. Kidney damage in many kidney diseases can be ascertained by the presence of albuminuria, defined as albumin-to-creatinine ratio >30 mg/g in two of three spot urine specimens.

-line 96, delete “Early”

-Please clearly describe the definition of early CKD (namely, stage 1-3a in this study).

-Add a sensitivity test for the association between re-definition of early CKD and uric acid and IHD.

-Please show the percentage of patients with CKD stage 1, 2, and 3a in this study.

2. “Hyperuricemia” rather than “uric acid level” may provide more clinical value. Thus, please add a sensitivity test for the association between hyperuricemia and early CKD and IHD.

-line 109, please provide the reference “of a threshold of ≥6.5...in women”.

3. How many people took anti-hyperuricemic drugs in this study? Did anti-hyperuricemic drugs effect the main result?

Discussion:

1. line 182-185: According to the results in table 3, the sentence “Uric acid was an independent indicator 182 of IHD irrespective of age, sex, body mass index, smoking status, alcohol intake, physical activity, mean arterial blood pressure, fasting plasma glucose, total cholesterol, hypertension medication, and dyslipidemia medication, particularly in early CKD patients.” should be revised to “Uric acid was an independent indicator of IHD irrespective of age, sex, body mass index, smoking status, alcohol intake, and physical activity.”

2. line 195: “over 4 years” of follow-up?? The abstract showed 2.3 year.

3. line 198-199: “Specifically, we showed a significant association even in less than stage 3a CKD.” The data was not present in the result!

Conclusion:

1. line 232-234: “The appropriate detection of both uric acid levels and CKD, even in the earlier stages, is essential to deter the development of IHD.” It may be powerful if  the authors can provide information on how high the uric acid level should be paid attention to.

Reviewer 2 Report

Dear authors, 

The paper tackles an interesting an often overlooked topic in nephrology. While there are several studies on this topic available in the literature, their results are not in agreement. 

This study included a large number of patient data derived from a database which provides quality data on a number of different laboratory and clinical parameters, allowing for adequate statistical analysis.

Several suggestions for improving the manuscript:

Abstract:

HRs of IHD for CKD and uric acid quartiles were 1.13 (95% CI, 0.86–1.48), 0.99 (95% CI, 0.55–1.77), and 1.65 (95% CI, 1.03–2.66) - which quartiles? Please denote the quartiles corresponding to the HRs displayed

Introduction

Hyperuricemia has been increasing over the decades - the incidence of hyperuricemia?

Furthermore, elevated uric acid levels are known to be associated with hypertension, diabetes, and metabolic syndrome, which are definitely risk factors for cardiovascular diseases - "known risk factors" instead of "definitely"

The early detection and correction of hyperuricemia is an important medical issue before the progression to comorbidity that induces a range of complications - "is an important medical issue that has to be addressed before the progression to comorbidity..."

Early CKD is an important public health problem before progression to ESRD - "is an important public health problem even before progression to advanced stages and ultimately ESRD"

In NHANES III, just 8% of patients knew their condition among all participants with a moderately decreased glomerular filtration rate - "only 8% of patients knew about their condition"

Conclusion

In summary, increased uric acid levels could be used to estimate the occurrence of IHD in patients with even less than stage 3a CKD. - "to estimate the occurrence of IHD in patients with CKD in stages even earlier than G3a"

The quality of the English language is satisfactory, with several minor changes needed, which are listed in the comments above. 

Round 2

Reviewer 1 Report

The authors have answered all of my questions and the paper has been greatly improved. Therefore, it can be accepted for publication.